# Protection and Alleviated Inflammation Induced by Virus-like Particle Vaccines Containing *Plasmodium berghei* MSP-8, MSP-9 and RAP1

**DOI:** 10.3390/vaccines10020203

**Published:** 2022-01-27

**Authors:** Su-Hwa Lee, Ki-Back Chu, Hae-Ji Kang, Fu-Shi Quan

**Affiliations:** 1Department of Medical Zoology, School of Medicine, Kyung Hee University, Seoul 02447, Korea; dltnghk228@khu.ac.kr (S.-H.L.); kbchu@khu.ac.kr (K.-B.C.); 2Medical Research Center for Bioreaction to Reactive Oxygen Species and Biomedical Science Institute, School of Medicine, Graduate School, Kyung Hee University, Seoul 02447, Korea; 3Department of Biomedical Science, Graduate School, Kyung Hee University, Seoul 02447, Korea; heajik0514@khu.ac.kr

**Keywords:** *Plasmodium berghei*, merozoite surface protein 8, merozoite surface protein 9, rhoptry-associated protein 1, virus-like particle, vaccine

## Abstract

Virus-like particles (VLP) are a highly efficient vaccine platform used to present multiple antigenic proteins. Merozoite surface protein 8 (MSP-8), 9 (MSP-9) and rhoptry-associated protein 1 (RAP1) of *Plasmodium berghei* are the important proteins in erythrocyte invasion and the replication of parasites. In this study, we generated three VLPs expressing MSP-8, MSP-9 or RAP1 together with influenza virus matrix protein M1 as a core protein, and the protection and alleviated inflammation induced by VLP immunization were investigated. Mice were immunized with a mixture of three VLPs, MSP-8, MSP-9 and RAP1, and challenge-infected with *P. berghei*. As a result, VLPs immunization elicited higher levels of *P. berghei* or VLPs-specific IgG antibody responses in the sera upon boost compared to that upon prime and naive. Upon challenge infection with *P. berghei*, higher levels of CD4+ T cell and memory B cell responses in the spleen were also found in VLPs-immunized mice compared to non-immunized control. Importantly, VLP immunization significantly alleviated inflammatory cytokine responses (TNF-α, IFN-γ) both in the sera and spleen. VLP vaccine immunization also assisted in diminishing the parasitic burden in the peripheral blood and prolonged the survival of immunized mice. These results indicated that a VLPs vaccine containing MSP-8, MSP-9 and RAP1 could be a vaccine candidate for *P. berghei* infection.

## 1. Introduction

Malaria caused by the *Plasmodium* parasite remains a serious disease threatening public health. In 2016, significant funds were invested to implement a malaria control and elimination program, which helped to reduce the burden of malaria disease [1]. However, the number of malaria cases worldwide in 2019 was 229 million, which has remained virtually unchanged over the past 4 years, and the number of deaths approached 409,000 [2]. Since malaria continues to cause enormous loss of life across world, there is a need for the introduction of a treatment that can prevent malaria, such as a vaccine [3]. Currently, the most advanced pre-erythrocytic stage malaria vaccine RTS,S, has demonstrated low to moderate efficacy (26–50%) in children who received four injections [4].

*P. berghei*, a rodent *Plasmodium* parasite, has been used extensively in vaccine development studies due to its structure, physiology and life cycle similar to that of human and other primate *Plasmodium* parasites [5,6]. Infection by the *Plasmodium* parasite can lead to death due to the severity of symptoms caused by inflammation [7,8,9]. Because the pathological symptoms of malaria disease begin to appear at the blood stage, we prioritized limiting the invasion of parasites in erythrocytes and the growth of invading parasites in vaccine design [10,11].

In our previous studies, we investigated the efficacy of virus-like particles (VLPs) vaccines by targeting the merozoite surface proteins (MSP). The MSP family forms the most abundant protein in the merozoite form of *Plasmodium*, which is considered an excellent target for vaccine development as an antigen that sufficiently stimulates the host’s immunity and is located in the parasitophorous vacuole (PV) during an erythrocytes infection of the host [12,13,14]. It was reported in our previous study that single merozoite surface protein 8 (MSP-8) or MSP-9 containing influenza VLPs vaccines sufficiently produce antibodies and significantly reduced the growth and proliferation of parasites [15,16]. However, the vaccine efficacies such as parasitemia and survival time need to be improved. This is thought to be due to the complex action of various proteins in the process of parasites invading host cells [12,17,18]. Vaccine design including multiple proteins involved in parasite invasion might be the preferred choice for inducing better vaccine efficacy.

*Plasmodium berghei* rhoptry-associated protein 1 (RAP1) is located in the rhoptry bulb organ of merozoites. Furthermore, RAP1 is formed into a complex with other rhoptry-associated proteins (RAP2 and RAP3) and is secreted from it, and the RAP complex is located in the parasitophorous vacuole (PV) formed after *Plasmodium* invades the host cell [19,20,21]. Additionally, RAP1-deficient *Plasmodium* parasites have been reported to have longer survival times in mice than normal *Plasmodium* parasites [22], indicating that RAP1 of the parasite is related to parasite toxicity.

In the study, we investigated the vaccine efficacy induced by multiple proteins MSP-8, MSP-9 and RAP1 containing VLPs. We found that a multiple-protein containing VLPs vaccine induces a parasite or VLPs-specific IgG antibody response. A higher population of CD4^+^ T and memory B cell responses in the spleen were observed in VLPs-immunized mice than non-immunized control upon challenge infection. Importantly, VLPs vaccination significantly reduced the parasitemia level and inflammatory cytokine (TNF-alpha, IFN-gamma) responses both in blood and spleen, and significantly prolonged survival time.

## 2. Materials and Methods

### 2.1. Animal Ethics Statement

All animal experiments were performed in strict accordance with the Kyung Hee University IACUC guidelines (permit number: KHSASP(SE)-19-188). Animals were housed in approved facilities with a 12 h day and night cycle, with easy access to food and water. To minimize animal suffering, mice were humanely euthanized upon reaching the humane intervention point which was determined to be at a weight loss exceeding 20% of initial bodyweight.

### 2.2. Experimental Reagents and Preparation of Animals, Cells, and Parasites

BALB/c mice (six-week-old, female) were purchased from NARA Biotech (Seoul, Korea). *Plasmodium berghei* ANKA strain 2.34 was serially maintained in mice through intraperitoneal passaging. Recombinant baculovirus (rBV) and VLPs were generated using *Spodoptera frugiperda* Sf9 cells. The Sf9 cells were cultured in spinner flasks with SF900II media purchased from Invitrogen (Carlsbad, USA). *P. berghei* whole antigens were prepared following the method previously described [15,16,23,24]. Blood samples were collected from *P. brerghei*-infected mice at 8 weeks post-infection via retro-orbital plexus puncture for serum acquisition. Monoclonal anti-M1 antibody and horseradish peroxidase (HRP)-conjugated secondary mouse IgG antibodies were purchased from Abcam (Cambridge, UK) and Southern Biotech (Birmingham, AL, USA), respectively. Fluorophore-conjugated antibodies were purchased from BD Bioscience (CA, USA) and used to perform flow cytometry.

### 2.3. Generation of P. berghei VLPs

For plasmid constructions, *P. berghei* merozoite surface protein 8 (MSP-8, codon-optimized), merozoite surface protein 9 (MSP-9) and influenza matrix protein 1 (M1) genes were cloned, as described previously [15,16,25]. Codon-optimized *P. berghei* rhoptry-associated protein 1 (RAP1) gene was acquired from GenScript (Piscataway, NJ, USA). 

The recombinant plasmid was transformed into DH10Bac competent cell. Colonies were screened and bacmid DNA from successful clonal construct was extracted using FavorPrep Gel Purification Kit (Favorgen, Cheshire, UK). MSP-8, MSP-9, RAP1, or influenza M1-expressing recombinant baculoviruses (rBVs) were prepared as previously described [26,27]. Three different VLPs, each expressing one of MSP-8, MSP-9, or RAP1 antigen were generated along with influenza M1 core protein as described previously [26,27]. After co-culturing Sf9 cells with the rBVs expressing the antigens of interest, culture media were collected at 3 days post-infection (dpi). Cells were pelleted by centrifuging for 1 h, 4 °C, 4800× *g*. VLPs present within the supernatants were purified and stored at 4 °C as described elsewhere [26,27]. 

### 2.4. VLP Characterization via Western Blot and Transmission Electron Microscopy

VLPs expressing the three *P. berghei* antigens were characterized using a transmission electron microscope (TEM) as described previously [15,16]. Purified RAP1 VLPs were negatively stained on the grid and observed under a TEM (JEOL 2100, JEOL USA, Inc.; Peabody, MA, USA) [28]. Twenty-five regions of the grid images were captured for analysis. Additionally, MSP-8, MSP-9 and RAP1 VLPs were characterized as indicated [15,16] by using 10% sodium dodecyl sulfate-polyacrylamide gel electrophoresis (SDS-PAGE) under denatured conditions. Proteins were separated based on their molecular weight and subsequently transferred onto a PVDF membrane (Millipore, MA, USA). Membranes were blocked with 5% skim milk prepared in TBST and then incubated with either the sera acquired from *P. berghei*-infected mice (1:500 dilution) or monoclonal mouse M1 antibody (1:2000 dilution) overnight at 4 °C. After washing, membranes were probed with HRP-conjugated mouse IgG secondary antibodies (1:3000) dilution for 1 h, RT.

### 2.5. Immunizing Mice with the VLPs and P. berghei Challenge Infection

Six-week-old female BALB/c mice were evenly distributed with appropriate control groups (*n* = 12 per group). The VLPs containing MSP-8, MSP-9, or RAP1 were mixed at a ratio of 1:1:1 (total 150 µg VLPs in 100 µL, 50 µg from each VLP). At weeks 0 and 4, a total of 150 µg VLPs comprising equal concentrations of MSP-8, MSP-9, and RAP1 VLPs were intramuscularly (IM) administered into mice. On week 8, which corresponds to 4 weeks after boost immunization, mice were intraperitoneally (IP) challenge-infected with 0.5% of *P. berghei* in 100 µL of PBS, which includes all forms of the parasite. At 7 dpi, six mice were randomly selected from each group and sacrificed for blood and spleen sample acquisition. The remaining mice in each group were monitored daily to record bodyweight changes and survival rates. 

### 2.6. Immune Cell Responses

To determine the parasite-specific IgG antibody response and VLP-specific IgG antibody response, mice sera were collected 3 weeks after prime and boost immunization, and the ELISA was performed as described previously [15]. Splenic T and B cells were analyzed by flow cytometry after lysing the red blood cells (RBCs) with the RBC lysis buffer purchased from Sigma-Aldrich (St. Louis, MO, USA). Both the CD4^+^ and CD8^+^ T cells were detected at 1 week post-challenge. Flow cytometry was performed as previously described [15,16]. Briefly, after FcR blocking for 15 min at 4 °C, splenocytes were stained with appropriate surface antibodies for 30 min, 4 °C. Cells were acquired and immune cell populations were analyzed using a BD Accuri C6 Flow Cytometer (BD Bioscience, San Jose, CA, USA).

### 2.7. Inflammatory Cytokine Analysis

At 1 week post-challenge, mice were sacrificed for blood and spleen sample acquisition. Spleens were homogenized in RPMI-1640 media purchased from Lonza (Basel, Switzerland), and supernatants from the individual mouse were carefully collected for cytokine detection as described previously [15,16]. Both TNF-α and IFN-γ cytokine productions were assessed using BD OptEIA Set (BD Biosciences, San Jose, CA, USA). All experiments were conducted following the manufacturer’s instructions. Optical density values of samples and standards were measured using a microplate reader. Standard curves were generated and cytokine concentrations of samples were calculated. 

### 2.8. Peripheral Blood Parasitemia Measurement

Retro-orbital plexus puncture was performed to acquire blood samples from *P. berghei*-infected mice. Blood samples were collected at 3, 7, 15, 22, 28, 34, 44, 50, and 56 days post-challenge infection. Briefly, 2 µL of blood samples were mixed with 100 µL of PBS pre-mixed with 500 U/mL of heparin. Prior to flow cytometry, samples were stained with 1 µL of SYBR Green I (Invitrogen, Waltham, MA, USA) and subsequently incubated for 30 min, 37 °C. Samples (10^4^ RBCs) were acquired using the BD Accuri C6 flow cytometer (BD Biosciences, San Jose, CA, USA) for analysis.

### 2.9. Statistical Analysis

Statistical significance between the groups was determined by performing a one-way analysis of variance (ANOVA) with Tukey’s post hoc test or Student’s t-test using PC-SAS 9.4 (SAS Institute, Cary, NC, USA). Data are expressed as mean ± SEM and asterisks were used to denote statistical significance. Differences between the groups were considered to be statistically significant for *P* < 0.05.

## 3. Results

### 3.1. Virus-like Particle Vaccines Were Generated

Codon-optimized RAP1 was synthesized by GenScript. The original and codon-optimized sequences, as well as the translated amino acid sequences, were tabulated (Appendix A). Successful integration of the RAP1 gene into the pFastBac vector was confirmed using restriction enzyme digestion with *EcoR*I/*Hind*III. Sf9 cells were co-infected with rBVs expressing *P. berghei* RAP1 and influenza M1 to generate RAP1 VLPs. The morphology of VLPs was determined by electron microscopy (Figure 1A–C). Furthermore, MSP-8, MSP-9 or RAP1 VLPs displayed a spherical morphology with spikes on their surfaces. A Western blot analysis was performed to confirm successful expressions of antigen components within the VLPs. Membranes probed with the anti-*P. berghei* polyclonal antibody acquired from infected mice reacted with the MSP-8, MSP9, and RAP1 antigens. Western blot bands for MSP-8 and MSP-9 VLPs were identical to ones reported previously [15,16]. Similarly, the presence of M1 proteins within the VLPs was confirmed using anti-influenza M1 antibody (Figure 1D–F).

### 3.2. VLPs Vaccine Elicited Parasite-Specific IgG Antibody Response

To evaluate the *P. berghei*-specific IgG antibody responses, sera were collected from immunized mice 3 weeks after each immunization (Figure 2). As shown in Figure 3A, potent antigen-specific IgG responses were detected after prime immunization, which was further enhanced following boost immunization. As expected, *P. berghei* antigen-specific response was not observed from the sera of naïve mice. Sera from immunized mice also reacted with VLPs (Figure 3B). These results indicated that vaccination with VLP induced a parasite-specific IgG antibody response.

### 3.3. Vaccination with VLPs Induced T Cells and Memory B Cell Responses

Mice were sacrificed after *P. berghei* challenge infection and splenocytes were acquired to assess splenic cellular immune responses elicited by the VLPs. Using fluorescence-activated cell sorting (FACS), splenic T cell (CD4^+^ and CD8^+^) and memory B cell (MB) responses were evaluated (Figure 4). T cells and memory B cells were analyzed following the procedure described previously [15,16]. Noticeably higher CD4^+^ T cell and MB cell populations were detected from spleens of immunized mice compared to non-immunized mice control (Figure 4A,C, ***** *P* < 0.05). These results indicate that VLPs immunization contributed to significantly increased CD4^+^ T cell and memory B cell populations.

### 3.4. VLPs Vaccination Significantly Reduced Pro-Inflammatory Cytokine Responses upon Challenge Infection with P. berghei

Serum and spleens were harvested at 7 dpi to evaluate the production of inflammatory cytokines TNF-α and IFN-γ. As shown in Figure 5, drastically diminished TNF-α levels (62%) were observed in the blood of immunized mice (A). Such reductions were also noticed for IFN-γ in both blood (92.6%) (B) and spleen (92.8%) (D). Compared to non-immunized naïve-challenged mice, both TNF-α and IFN-γ levels in the immunized mice were significantly less (***** *P* < 0.05, ****** *P* < 0.01). These results indicated that VLP immunization induced significant reductions of inflammatory cytokines in blood and spleen.

### 3.5. VLPs Vaccination Led to Significantly Reduced Parasitemia, Delayed Bodyweight Loss, and Prolonged Survival

Blood samples were drawn from all groups of mice after *P. berghei* infection to evaluate the parasitic burden in mice (Figure 6A). At 28 days post-challenge infection, mice showed significant reductions in parasitemia were observed from the blood of vaccinated mice (680 parasites) compared to those of unimmunized controls (5800 parasites, ****** *P* < 0.01). Mice immunized with VLPs maintained their normal weight for 48 days, while the non-immunized mice control began to experience reduced body weight from day 4 post-challenge infection (Figure 6B). Immunized mice survived much longer than non-immunized control mice (Figure 6C). These results indicated that VLP immunization contributed to significant reductions in parasitemia level, body weight loss and survival.

## 4. Discussion

In this study, we investigated the efficacy induced by multiple proteins (MSP-8, MSP-9 and RAP1) containing virus-like particles (VLPs) vaccine for *Plasmodium berghei* parasite infection. VLPs vaccines expressing MSP-8, MSP-9 or AMA1 are known to induce significantly reduced parasitemia and inflammatory cytokines [15,16,29]. However, that mice still show body weight loss and finally die indicates that an improvement of vaccine efficacy is needed. In the current study, multiple proteins of parasites involved in parasite invasion and growth were selected for VLP generation using the baculovirus expression system. We found that vaccination with multiple proteins MSP-8, MSP-9 and RAP1 containing virus-like particles (VLPs) significantly improved vaccine efficacy by showing significantly reduced parasitemia and inflammatory cytokines. Currently, the most advanced malaria vaccine is RTS,S, targeting the pre-erythrocytic stage of the parasite. However, in the current study, using MSP-8, MSP-9, and RAP1, we generated targets blood-stage parasites. 

RAP1 is known to form a parasitophorous vacuole (PV), which is necessary for parasite growth in host cells and contributes to parasite toxicity [19,20,21,22]. Since MSP-8 and MSP-9 in VLPs immunization studies showed limited protection, we hypothesized that multiple VLPs vaccines containing MSP-8 and MSP-9, as well as RAP1, could reach higher levels of protection against *P. berghei* infection. As expected, vaccination with VLPs containing multiple proteins MSP-8, MSP-9 and RAP1 showed a significant reduction (nine-fold) of parasitemia in the blood compared to single MSP-8 or MSP-9 VLPs vaccination. 

In our previous studies, MSP-8 or MSP-9 expressing VLPs immunization limited the production of pro-inflammatory cytokines, which contributed to the survival and reduced parasite burden in immunized mice [15,16]. In agreement with these results, in the current study, inflammatory cytokine IFN-γ was found to be significantly decreased (90-fold) in multiple protein VLPs immunization compared to those from single MSP-8 or MSP-9 VLPs immunizations.

Severe pathology in mice infected with *Plasmodium berghei* is known to overproduce pro-inflammatory cytokines IFN-γ and TNF-α [7,8]. Additionally, TNF-α plays an important role in the pathogenesis of malaria disease, and the local accumulation and activation of macrophages can lead to severe lesions [30]. It has been reported that profound levels of pro-inflammatory cytokines, such as TNF-α and IL-6 are associated with severe malaria [31]. The increased malaria parasite loads in the blood stage exacerbate the pro-inflammatory response associated with severe malaria [15,31,32,33]. Our previous studies indicated that *P. berghei* is highly pathogenic and uniformly lethal, resulting in severe pathology in mice upon IP route infection [15,16]. The results are consistent with those reported by others [34]. The contribution of inflammatory cytokine responses seems to be much more complex and needs further dissection. Further studies are required to elucidate the mechanisms that underlie the dynamic of inflammatory cytokine response related to protection or pathology. During plasmodium parasite infection, the spleen of the host strengthens immune function and filtration to accelerate the removal of plasmodium-infected red blood cells [35]. In the current study, we found that multiple protein-containing VLPs immunization significantly reduced proinflammatory cytokines IFN-γ and TNF-α both in spleen and blood, which might contribute to the significantly reduced body weight loss and prolonged survival. Consistent with these results, we found that spleen size (weight) in the VLPs immunized mice was smaller than those in non-immunized naïve challenge control (data not shown), indicating that lower pathogenesis occurred by showing lower inflammatory cytokine responses induced by vaccination with VLPs.

Mouse survival and body weight changes are important indicators for the evaluation of vaccine efficacy. Compared to single VLPs vaccines, multiple proteins containing VLPs vaccine showed significantly improved body weight loss and survival. Individual AMA1, MSP-8 or MSP-9 expressing VLPs immunization showed significant increases in parasite-specific IgG antibody responses, T cell responses, and B cell responses, correlating with significant reductions of parasitemia and prolonged survival time [15,16,29].

## 5. Conclusions

In conclusion, we investigated the vaccine efficacy induced by VLPs vaccine containing *P. berghei* MSP-8, MSP-9 and RAP1 in mice. We found that VLPs vaccination led to higher levels of *P. berghei*-specific IgG antibody response, CD4^+^ T cell and memory B cell responses. Vaccination with VLPs significantly reduced the pro-inflammatory cytokine responses, parasitemia and body weight loss. These results indicated that multiple protein-containing VLPs could be an effective vaccine candidate for malaria parasite infections.

## Figures and Tables

**Figure 1 vaccines-10-00203-f001:**
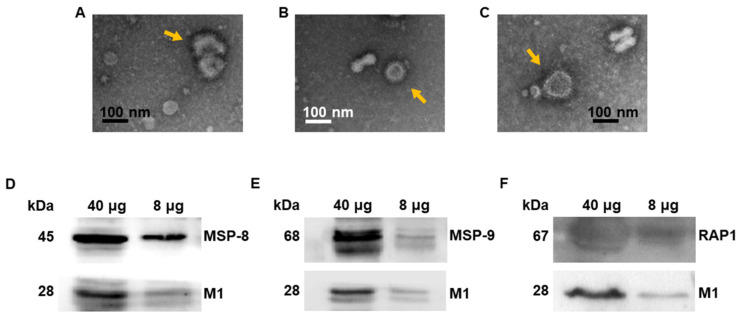
VLPs characterization. MSP-8 (**A**), MSP-9 (**B**) and RAP1 (**C**) VLP morphologies were visualized under TEM. MSP-8 (**D**), MSP-9 (**E**) and RAP1 (**F**) VLPs were loaded (8, 40 µg) for SDS-PAGE, and bands corresponding to each antigen were observed using either the polyclonal *P. berghei* antibody or the monoclonal M1 antibody.

**Figure 2 vaccines-10-00203-f002:**
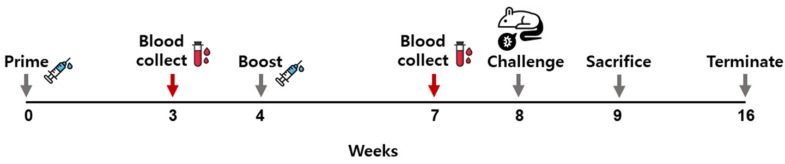
Experimental schedule. Mice were immunized twice with the combined VLP vaccines expressing MSP-8, MSP-9, and RAP1 of *P. berghei* at 4 weeks intervals, and blood samples were drawn 3 weeks after each immunization. Challenge infection was performed 4 weeks after boost immunization and half of the mice in each group were sacrificed at 1 week post-challenge.

**Figure 3 vaccines-10-00203-f003:**
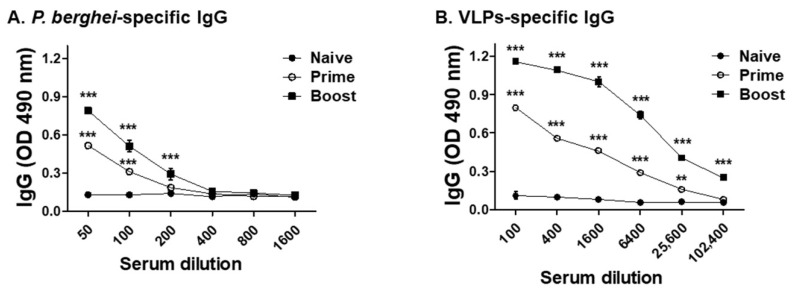
IgG antibody response. Mice were intramuscularly immunized twice with the mixture of MSP-8, MSP-9 and RAP1 VLPs. Sera (*n* = 6) were collected at regular intervals after each immunization with the VLPs. VLPs immunization induced IgG antibodies reacted with *P. berghei* Ag (**A**) and VLPs (**B**) upon prime and boost compared to naïve sera (** *P* < 0.01, *** *P* < 0.001).

**Figure 4 vaccines-10-00203-f004:**
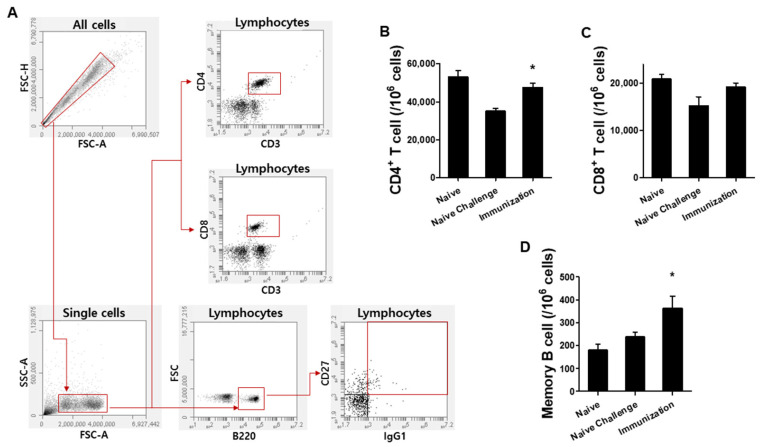
Immune cell responses. Gating strategy for CD4^+^, CD8^+^ T cells and memory B cells (**A**). Numbers of CD4^+^ (**B**) and CD8^+^ (**C**) T cells and memory B cells (**D**) were analyzed by flow cytometry after *P. berghei* challenge infection. Higher numbers of CD4^+^ T cells (**B**, * *P* < 0.05) and memory B cells (**D**, * *P* < 0.05) were observed from the spleens of VLP immunization group compared to non-immunized control (*n* = 6).

**Figure 5 vaccines-10-00203-f005:**
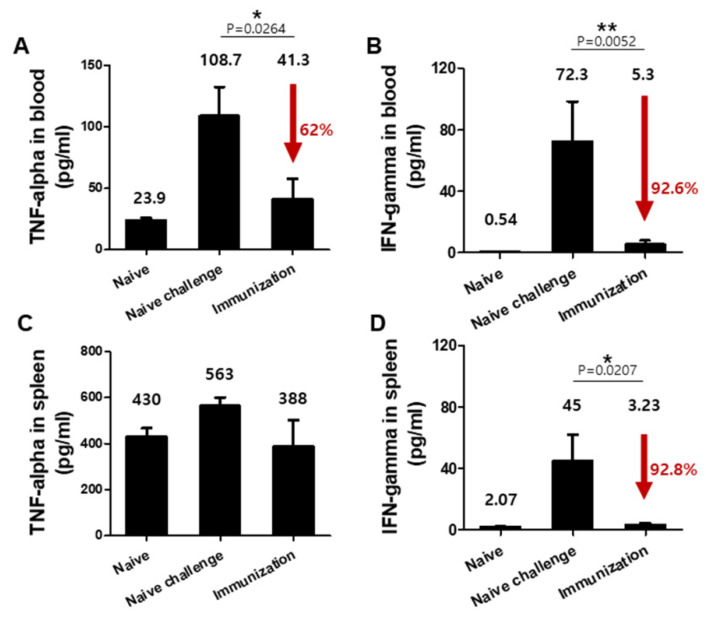
Pro-inflammatory cytokines (TNF-α and IFN-γ) response. To determine the levels of inflammatory cytokines TNF-α (**A**,**C**) and IFN-γ (**B**,**D**) upon challenge infection, mouse blood (*n* = 6) and spleen (*n* = 6) was collected at 7 dpi. Higher levels of TNF-α (**A**), * *P* < 0.05 and IFN-γ (**B**), ** *P* < 0.01 in the blood and IFN-γ (**D**), * *P* < 0.05 in the spleen were detected in the non-immunized control (Naïve Challenge) compared to VLPs immunized mice (Immunization).

**Figure 6 vaccines-10-00203-f006:**
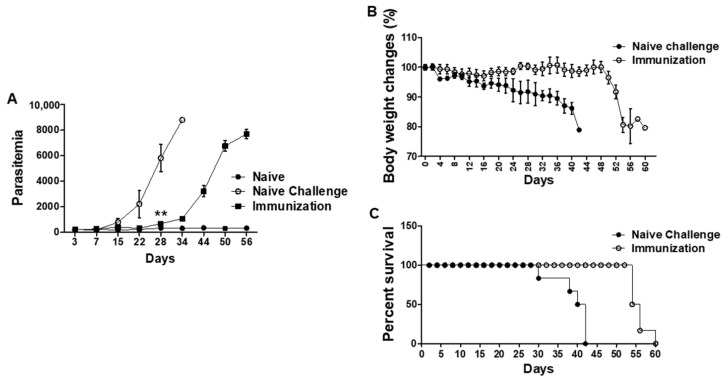
Parasitemia and survival. RBCs of *P. berghei* challenge-infected mice were collected for parasite enumeration via flow cytometry (**A**), ** *P* < 0.01. Body weight changes and survival rates of mice were determined at two days interval after *P. berghei* infection (**B**,**C**). The error bars represent SEM.

## Data Availability

Data supporting the findings of this study are contained within the article.

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
