# Peer review of "Protection and Alleviated Inflammation Induced by Virus-like Particle Vaccines Containing *Plasmodium berghei* MSP-8, MSP-9 and RAP1"

_vaccines, 2022, doi:10.3390/vaccines10020203_

Round 1

Reviewer 1 Report

This is a very important research because these results indicated that a
VLPs vaccine containing MSP-8, MSP-9 and RAP1 could be a vaccine candidate for P. berghei infection.

Author Response

Response: Thank you so much!

Reviewer 2 Report

The manuscript presented by Lee and colleagues uses a VLP platform for the development of multivalent vaccines in the context of malaria. The data presented are interesting and present initial data for the characterization of the immunization process. However, in a next step, the authors need to carry out other proofs of concept to understand the mechanism of immunity induced by the presented methodology.

Author Response

Response: Thank you for the reviewer's comments. We would like to carry out other proofs of concept to understand the mechanism of immunity induced by our VLP vaccines.   

Reviewer 3 Report

The work done by Lee and colleagues is of importance to the fight against malaria.
In the introduction, the authors should refocus their study in relation to recent advances in malaria control, notably the licensing of the RTS S vaccine. What contribution can their study make in this context? This should also be discussed in the discussion section.

The authors have shown that the combination of three antigens seems to be beneficial in the immune response, survival and growth of mice. In their previous studies they showed that the use of MSP 8 and MSP 9 needed to be improved, this is the main reason why they included RAP1 antigen. However, Why don't they just measure the effect of RAP1 on the development of the immune response to see if it is necessary to combine these three antigens?

The methodology is the main problem of this study. The authors do not take the trouble to describe their procedures correctly and they refer too often to studies already carried out.
The immunization part is not clear enough between what is described in the methodology and what is presented in the results. 
In the results section, there are mainly three groups: naive, prime and boost. However, in the methodology, it is difficult to perceive these different groups. For more clarity, the authors should clearly define these groups in the methodology section to allow a better understanding of the results. The different stages of the immunization protocol should be presented more clearly. A diagram showing the different groups and different challenges with the types of antigens could be helpful. This will allow a better understanding of part 3.2 of the IgG response results and to know exactly which post-immunization period each graph corresponds to. Also, in Figure 2 in order to compare the specific IgG response to P.berghei and VLP, the authors did not give any indication on the statistical analyses between the different groups and conditions. What is the interest of using sera dilutions to assess the IgG response? In any case, mice challenged with P.berghei ag produce less antibodies than those exposed to VLP only, so what is the relevance of the IgG response detected with P.berghei ag compared to that with VLP?
It is essential to describe in the methodology the procedure used to measure IgG production. The methodology used to measure cytokine production is not well described. How did the authors determine the thresholds of positivity?

Minors remarks

 Page 3 Line 113: to move in results section
Page 3 Line 124  150 micro g of VLP were injected to each group, what is really the concentration of the different antigen?

Page 3 Lines 126-127Concerning the challenge with parasites, the authors should indicate which stage (form) (trophozoite schizont?) was used.
Page 4 section 2.9 For the measurement of parasitemia, the authors should specify at which period each measurement was made

Figure 3A: it would be interesting if the authors give more details on how the the gating was made.

Author Response

Reviewer 3

The work done by Lee and colleagues is of importance to the fight against malaria.
In the introduction, the authors should refocus their study in relation to recent advances in malaria control, notably the licensing of the RTS S vaccine. What contribution can their study make in this context? This should also be discussed in the discussion section.

Response: The most advanced malaria vaccine RTS,S has been described in the introduction section (lines 38-40) and Discussion section (lines 265-268).     

The authors have shown that the combination of three antigens seems to be beneficial in the immune response, survival and growth of mice. In their previous studies they showed that the use of MSP 8 and MSP 9 needed to be improved, this is the main reason why they included RAP1 antigen. However, Why don't they just measure the effect of RAP1 on the development of the immune response to see if it is necessary to combine these three antigens?

Response: In our previous study, single MSP 8 or MSP 9-expressing VLP vaccines have shown limited protection. Also, a longitudinal study involving patients naturally infected with P. falciparum revealed that anti-RAP1 antibody responses in these individuals were very short-lived (Infect Immun 1999; 67(6):2975-2985). Thus, we do not expect that individual RAP1 can show better protection than MSP 8 or MSP 9.     

The methodology is the main problem of this study. The authors do not take the trouble to describe their procedures correctly and they refer too often to studies already carried out.
The immunization part is not clear enough between what is described in the methodology and what is presented in the results. 

Response: More details on the experimental procedure has been added in the M&M (lines 125-126, 129-130, 135-137, 147-148, 155-156 ).

In the results section, there are mainly three groups: naive, prime and boost. However, in the methodology, it is difficult to perceive these different groups. For more clarity, the authors should clearly define these groups in the methodology section to allow a better understanding of the results. The different stages of the immunization protocol should be presented more clearly. A diagram showing the different groups and different challenges with the types of antigens could be helpful. This will allow a better understanding of part 3.2 of the IgG response results and to know exactly which post-immunization period each graph corresponds to. Also, in Figure 2 in order to compare the specific IgG response to P.berghei and VLP, the authors did not give any indication on the statistical analyses between the different groups and conditions. What is the interest of using sera dilutions to assess the IgG response? In any case, mice challenged with P.berghei ag produce less antibodies than those exposed to VLP only, so what is the relevance of the IgG response detected with P.berghei ag compared to that with VLP?
It is essential to describe in the methodology the procedure used to measure IgG production. The methodology used to measure cytokine production is not well described. How did the authors determine the thresholds of positivity?

Response: A diagram has been provided as Figure 2 (lines 195-200), so that other Figure numbers are corrected. Statistical analysis has been indicated for Figure 2 which is now Figure 3 (lines 205-206). Methodology for IgG and cytokine responses have been newly described (lines 135-137 and 147-148). 

The positivity was determined by comparing OD values with Naive for IgG antibody response, or by comparing pg/ml with Naive challenge for cytokine production. If the statistical analysis was significant between Naive and Boost for IgG production or between immunization and Naive challenge for cytokine response, they are positive.

Minors remarks

Page 3 Line 113: to move in results section

Response: It has been moved (lines 178-179)

Page 3 Line 124  150 micro g of VLP were injected to each group, what is really the concentration of the different antigen?

Response: The concentration of different VLPs has been indicated (lines 125-126). 

Page 3 Lines 126-127Concerning the challenge with parasites, the authors should indicate which stage (form) (trophozoite schizont?) was used.

Response: All stages (forms) of parasites in the blood stage were included for challenge infection (line 130).   

Page 4 section 2.9 For the measurement of parasitemia, the authors should specify at which period each measurement was made

Response: As indicated in Figure 6A, blood samples were collected on days 3, 7, 15, 22, 28, 34, 44, 50, and 56 post-challenge infection. This information has been added (lines 155-156). 

Figure 3A: it would be interesting if the authors give more details on how the the gating was made.

Response: No more details can be provided.

Reviewer 4 Report

See attached.

Author Response

Reviewer 4

Lee et al. Protection and alleviated inflammation induced by virus-like particle vaccines containing Plasmodium berghei MSP-8, MSP-9 and RAP1 Malaria is a mosquito-borne disease caused by Plasmodium parasites. Malaria remains a public health burden with 229 million new infections and approximately 0.5 million deaths worldwide in 2019. The approval of the world’s first malaria vaccine (RTS,S/AS01, trade name Mosquirix) in 2015 by the European Medicine Agency (EMA) and the World Health Organization (WHO) endorsement for broad use of the licensed vaccine in young children last year represent a paradigm shift in malaria control. Nonetheless, the Mosquirix vaccine is far from perfect as it requires a 4-dose regime and only provide partial protection against malaria. Novel or established vaccine platforms, such as mRNA, virus-like particle (VLP), viral vector technologies, may improve malaria vaccines efficacy while reducing the timeline to complete the vaccination schedule. Lee et al. developed a mixture of VLPs as vaccine in which three antigens (MSP-8, MSP-9 and RAP1) from P. berghei, a rodent Plasmodium parasite, is expressed. The authors characterized the VLP by electron microscopy and western blot. As expected, mice immunization with this VLP induced both humoral and cellular immunity. Upon challenge, immunized mice showed reduced levels of pro-inflammatory cytokines. Although vaccine delayed parasitemia and body weight loss, immunized mice still succumbed to Plasmodium infection. While I believe the manuscript is not suitable for publication based on the presented data, I would like to commend the authors on the well-written manuscript.

1) A major flaw in the present study is the lack of important controls. There is no control VLP with an unrelated antigen in the immunization experiments. What evidence do the authors have that protection is mediated by the three antigens (MSP-8, MSP-9 and RAP1) in the VLP? The authors showed no evidence that antibodies induced by vaccination are functional against the parasite. There are no experiments assessing the specificity of B and T cells, such as ELISPOT, antigen-specific staining in flow cytometry, or killing assay ex vivo.

Response: Previously, we have investigated the protective immunity induced by vaccine antigen-negative M1 VLPs, which cannot induce protective immunity (J Virol 2007; 81(7): 3514–3524 ; Pathogens 2021; 10(10):1291). We assessed antigen-specific T cell responses (Figure 4B, C) and memory B cell responses (Figure 4D) by flow cytometry. We have provided the parasitemia, body weight change, and survival, rather than performing killing assay as previously published (Vaccines 2020; 8(3): 428).         

2) Line 79: What surgery were performed in the animals?

Response: It has been corrected (lines 79-81).

3) Line 127: What is the infectious unit for the challenge? Can the authors clarify what the following statement means? “0.5%/ 100 μl of P. berghei in PBS”. 0.5% of what exactly?

Response: It has been corrected (lines 129-130). The parasite concentration is 0.5%.  

4) Figure 1D-F: The authors need to upload the full image of the western blot as a supplementary figure.

Response: Full image of the western has been provided in the supplementary materials. 

5) Why the discrepancy between P. berghei-specific IgG vs. VLPs-specific IgG antibody responses? How were IgG antibody responses measured? How do the authors normalize antigen amount for the ELISA? I did not find in the methods section.

Response: The coating antigens for ELISA between P. berghei-specific IgG and VLPs-specific IgG antibody response detection are different. That is why they showed the discrepancy. Protein assay for P. berghei antigen or VLP antigen for ELISA was carrried out and 4ug/ml concentration was used for coating into the ELISA plate. The methodology for P. berghei-specific IgG vs. VLPs-specific IgG antibody responses have been added in the Method section (lines 135 -137).  

6) Figure 3. How do the authors explain the robust cellular immune response in naïve mice group compared to naïve challenged or immunization groups? Would it not be more relevant to measure cellular immune response post vaccination and not post challenge?

Response: During the blood stage of malaria infection, the spleen was enlarged with a great influx of non-B and non-T cells (Infect Immun, 2000; 68(3):1485-1490). Thus, the percentage of CD4+ and CD8+ T cells in naïve challenged or immunized groups in our study were lower compared to Naïve. That is why naïve mice showed higher cellular responses compared to naïve challenged mice. It would be more relevant to measure immune response post-challenge.

7) Figure 4C. Is it expected to see this level of TNF-α in naïve mice? It seems too high for a “naïve” mouse in my view.

Response: Although the level of TNF-a from Naive looked high, it is significantly lower compared to Naive challenge.

8) Line 233: “VLPs vaccination led to significantly reduced parasitemia, body weight loss and survival”. I must disagree with this statement. There is no reduced parasitemia or body weight loss. Based on figure 5, what it is observed is a delay in both parasitemia and body weight loss. And while this delay may prolong the survival curve, immunized mice still succumbed to the challenge.

Response: As seen in figure, for parasitemia, we focused on day 28 post-challenge infection, which has been newly added (line 242). At day 28 post-challenge, vaccinated mice showed 6.8% of parasitemia reduction whereas unimmunized control showed 58%, which is statistically significant (P < 0.01) as indicated. To be clear, we have newly modified the description regarding parasitemia, body weight, and survival (lines 239-240).    

9) Figure 5A: Why parasitemia is presented as a percentage and not with absolute parasite count? How is the percentage calculated? The details are missing from the methods section. Why not present the absolute numbers or both the percentage and absolute numbers?

Response: Parasitemia was measured by flow cytometry analysis as described in the Material section, in which percentage data was provided.    

10) Figure 5C: How the authors explain the discrepancy in the survival curve presented in the current manuscript compared to the group previous work (reference 14)? I am specifically referring to the naïve challenge group (0% survival by day 19 in previous work compared to 100 by day 29 in the present manuscript).

Response: In the previous work (reference 14, now reference 15), mice were challenge infected with higher dose of parasites (1%), while in the current study, mice were challenge infected with 0.5% of parasites.

11) Lines 280-281: “Since IFN-γ has shown to be either protective or immunopathological in malaria”. This sentence seems to be lost there. Is it a punctuation typo or something else?

Response: This sentence has been removed.

Round 2

Reviewer 3 Report

The authors have taken care to provide answers to all questions and suggestions. This has improved the quality of the manuscript and made it more understandable.

Author Response

Thank you so much!

Reviewer 4 Report

I am mostly satisfied with the changes made by the authors. The only point we still disagree with refers to the plotting of the percentage of cells vs. absolute cell count. If enhanced cellularity (of non-B and non-T cells) in the spleen is observed during the blood stage of malaria infection, it makes even more sense to plot absolute numbers instead of percentage. The same is true for parasitemia. From flow cytometry experiments, you can obtain both cell count as well as the percentage of the cells of a given population. It is not that hard to get the information from the authors' flow cytometry data.

Author Response

We have revised Figures 4 and 6 as reviewer commented, in which absolute cell counts were provided and related descriptions were corrected (lines 160, 221, 222, 245).